# Clinical Use of an Order Protocol for Distress in Pediatric Palliative Care

**DOI:** 10.3390/healthcare7010003

**Published:** 2019-01-02

**Authors:** Marc-Antoine Marquis, Lysanne Daoust, Edith Villeneuve, Thierry Ducruet, Nago Humbert, France Gauvin

**Affiliations:** 1Division of General Pediatrics, Department of Pediatrics, Sainte-Justine Hospital, Montréal, QC H3T 1C5, Canada; 2Pediatric Palliative Care Unit, Department of Pediatrics, Sainte-Justine Hospital, Montréal, QC H3T 1C5, Canada; lysanne.daoust.hsj@ssss.gouv.qc.ca (L.D.); nago.humbert@umontreal.ca (N.H.); france.gauvin@umontreal.ca (F.G.); 3Division of Pediatric Anesthesiology, Department of Pediatrics, Sainte-Justine Hospital, Montréal, QC H3T 1C5, Canada; edith.villeneuve.hsj@ssss.gouv.qc.ca; 4Applied Clinical Research Unit, Department of Pediatrics, Sainte-Justine Hospital, Montréal, QC H3T 1C5, Canada; thierry.ducruet@gmail.com

**Keywords:** pediatric palliative care, dyspnea, pain control, order protocol, end of life care

## Abstract

Several children receiving palliative care experience dyspnea and pain. An order protocol for distress (OPD) is available at Sainte-Justine Hospital, aimed at alleviating respiratory distress, pain and anxiety in pediatric palliative care patients. This study evaluates the clinical use of the OPD at Sainte-Justine Hospital, through a retrospective chart review of all patients for whom the OPD was prescribed between September 2009 and September 2012. Effectiveness of the OPD was assessed using chart documentation of the patient’s symptoms, or the modified Borg scale. Safety of the OPD was evaluated by measuring the time between administration of the first medication and the patient’s death, and clinical evolution of the patient as recorded in the chart. One hundred and four (104) patients were included in the study. The OPD was administered at least once to 78 (75%) patients. A total of 350 episodes of administration occurred, mainly for respiratory distress (89%). Relief was provided in 90% of cases. The interval between administration of the first protocol and death was 17 h; the interval was longer in children with cancer compared to other illnesses (*p* = 0.02). Data from this study support the effectiveness and safety of using an OPD for children receiving palliative care.

## 1. Introduction

Several children in palliative care experience dyspnea and pain, throughout their illness and at the end of their lives [1,2,3,4]. Relief of these symptoms is a cornerstone of palliative care [5], and a priority for research in the field [6,7].

Use of standardized order templates for medication administration is a strategy employed to simplify the prescription of pharmacological therapies, including opioids and benzodiazepines, and to optimize their diligent administration for symptom relief [8]. Starting in 2009, an order protocol for distress (respiratory distress, acute pain crisis or anxiety) was established for pediatric palliative care patients at Sainte-Justine Hospital. The order protocol for distress (OPD) is a prescription template comprising opioids, benzodiazepines and anticholinergic drugs. The attending physician completes the prescription whenever indicated, based on evolution of the patient’s illness. The nurse can administer the OPD whenever the patient presents symptoms of acute distress; the physician must then be notified that the OPD has been initiated.

The present study was designed to assess the clinical use of the OPD in pediatric palliative care patients at Sainte-Justine Hospital, with emphasis on the protocol’s effectiveness and safety.

## 2. Materials and Methods

This descriptive epidemiological study was based on a retrospective chart review of all patients for whom the OPD was prescribed between September 2009 and September 2012. Patients were either hospitalized at Sainte-Justine Hospital, at the long-term care facility Marie-Enfant Rehabilitation Center, staying at Le Phare hospice, or followed at home.

The following elements were retrieved from the charts and recorded: demographic data (gender, age) and diagnosis; data pertaining to the trajectory of patients enrolled in the study (date of first consultation with palliative care services, date of prescription of the OPD, date and time of OPD initiation, date and time of death); and data related to administration of the protocol (route of administration—intravenous or subcutaneous, indication stated for initiating the protocol, person who administered the medication, actual medication received by the patient, location at which the medication was administered, pain evaluation in nursing or physician’s notes, modified Borg scale).

### 2.1. Details of the OPD

The attending physician and the pediatric palliative care team can prescribe the OPD to a child when there are reasons to expect he or she might experience respiratory distress, acute pain or anxiety in the setting of a life-threatening illness. Criteria for OPD prescription are presented in Table 1.

The order protocol combines a benzodiazepine and an opioid, sometimes with an anticholinergic agent. A standardized prescription form is readily available via the hospital pharmacy’s private network. It is filled out and signed by a physician, who can refer to suggested usual recommended dosing (intravenous or subcutaneous) included on the form. Medication is then prepared and made available in a locked cabinet on the care unit, accessible to the patient’s bedside nurse.

The bedside nurse can initiate the OPD whenever the patient presents acute distress not relieved by his or her usual medication. Once the OPD is initiated, the nurse must notify the physician, who is required to come to the patient’s bedside to evaluate the cause of distress, adjust medication if necessary, and provide support to the patient and family. The nurse can repeat administration of the OPD medication up to 2 times, every 15 min as needed, until the physician is present.

### 2.2. Outcomes

Effectiveness of the OPD was assessed using chart documentation of the patient’s symptoms. The intervention was classified as having provided effective relief, partial relief or no relief of symptoms, according to nursing or physician notes or an assessment scale (before and 10 min after the protocol medication was administered). Given there is no validated scale for dyspnea in the pediatric palliative care population [9], the modified Borg Scale (1 to 10) was used in this study [10]. To ensure uniformity, nurses were responsible for recording the score in every case; the same nurse would rate the patient’s dyspnea both before and after administration of the OPD medications. We considered effective relief of symptoms was achieved if there was an improvement of at least 3 points between both measurements; partial relief was understood as an improvement of 1 or 2 points between both measurements; we recorded no improvement if the score was the same or worse after administration of the OPD.

Safety of the OPD was evaluated by measuring the time between administration of the first medication and the patient’s death, as well as the clinical evolution of the patient as recorded by the nurse and physician in the patient’s chart.

### 2.3. Data Management

A case report form was developed and pretested by two investigators (F.G. and L.D.). Case report forms were filled out with data that were extracted from hospital charts by one investigator (F.G.). All case report forms were reviewed a second time (F.G.) to minimize errors and data were entered into an Excel database by the same investigator (F.G.). Data management fulfilled requirements of standard good clinical data management practice, and all data were treated anonymously.

### 2.4. Statistical Analysis

Frequencies, proportion, mean, standard deviation, median and range where used for descriptive statistics. Student t test or non-parametric Wilcoxon test were applied when appropriate. A *p*-value of 0.05 was used to define the threshold of statistical significance. All analyses were performed using the SAS Statistical Software V9.3 (SAS Institute, Cary, NC, USA).

This study was approved by the Institutional Review Board of Sainte-Justine Hospital and was funded by the “Centre d’excellence en soins palliatifs pédiatriques du CHU Sainte-Justine”.

## 3. Results

### 3.1. Patients’ Characteristics

Over a 3-year period the OPD was prescribed to 107 patients (Figure 1). Three patients were excluded because they were lost to follow-up after being transferred to another hospital; thus 104 were analyzed in our study. Characteristics of this cohort are detailed in Table 2. Patients were mostly girls (57%) and median age was 7.1 years (range: 14 days to 22 years). The main diagnoses were cancer (48% of cases), encephalopathy/anoxic brain injury (23%) and genetic disorders (11%).

### 3.2. OPD Characteristics and Administration

The protocol was administered at least once to 78 (75%) patients. A total of 350 episodes of OPD administration occurred in this cohort. The number of OPD received by individual patients is detailed in Figure 2. The median number of OPDs received was 3 (range: 1 to 30). Half of the repeated OPDs were administered within 120 min of a previous administration; twenty-five percent were administered within 30 min (Figure 3).

Ten patients (13%) received the protocol more than 10 times. Among these patients, 5 had severe neurological conditions with varying degrees of epilepsy, 3 had solid malignant tumors and 2 had spinal muscular atrophy. The median time between two OPD administrations in these patients was 72 min (range: 10 min to 71 days).

Characteristics of OPD administration are detailed in Table 3. Midazolam was prescribed in 98% of cases. Morphine was the preferred opioid (75% of cases), whereas hydromorphone was used in 24% of OPD. Glycopyrrolate and scopolamine were ordered less frequently, respectively 25% and 7% of the time.

The OPD was administered at Sainte-Justine Hospital in 68% of cases. Initiation of the OPD at Le Phare Hospice (21%), Marie-Enfant Rehabilitation Center (7%) and at home (3%) were less frequent. In most instances (75%), the nurse was alone in initiating the protocol. Indication to initiate the OPD was respiratory distress in 89% of episodes (*n* = 310), gasping representing 12% of all episodes. In patients who received the OPD more than 10 times, respiratory distress was the most common indication for administration; gasping was stated more frequently as the reason for administration (20% of cases).

In patients who did not receive an OPD, the reasons stated were: absence of distress (61%), patient relieved by other medication (12%), refusal of the family or patient (12%), refusal of the physician or resident (8%), or not specified (7%).

### 3.3. Intervals

The median interval between the initial palliative care consultation and OPD prescription was 34 days (range: 0 days to 2.4 years). The median interval between the prescription of the OPD and its first administration was 3.5 days (range: 0 days to 17 months). When comparing patients according to their underlying illness, the interval between prescription of the OPD and its first initiation was longer in children with cancer than any other disease (*p* = 0.02). There was no statistical difference in children with cancer as compared to other children with regards to time between the first palliative care consultation and prescription of the OPD, or between prescription of the OPD and death.

### 3.4. Effectiveness of the OPD for Symptom Management

Effectiveness of the protocol to relieve symptoms is presented in Table 4. In 77 episodes of OPD administration the level of relief was unknown as it was not documented in the chart. Among the 273 episodes of OPD administration where pertinent data were available, relief of symptoms was obtained 90% of the time. Within this group, 68% of OPD resulted in effective relief and 22% in a partial relief. In 10% of episodes the protocol did not seem to relieve the symptoms.

### 3.5. Safety of OPD Administration

#### 3.5.1. Interval between OPD Administration and Death

The median interval between administration of the first protocol and death was 17 h (range: 1 min to 6 months). Twenty-five percent of patients died within 4 h of their first OPD administration. Forty percent died more than 24 h after initiation of the OPD (Figure 4). There was no statistical difference in children with cancer as compared to other children with regards to time between administration of the OPD and death.

#### 3.5.2. Clinical Evolution after OPD Administration

Detailed review of charts provided no evidence that the OPD had precipitated death or caused harm to the patient.

At the end of the study, ten patients were still alive and at home; four of them had received the protocol at least once.

## 4. Discussion

The aim of OPD is to provide rapid and effective relief of pain, anxiety, and respiratory distress for children with a life-threatening illness receiving palliative care, including patients at the end of life. In our study, 75% of children for whom the OPD was prescribed received it at least once.

Results of our study show that the OPD provided relief of symptoms in most patients. OPD administration was effective in 90% of cases in which assessment data was available. Even if we were to assume the worst-case scenario where all missing data (77 episodes) were added to the 28 documented non-responders, a 70% improvement rate would remain. These results are comforting considering the level of suffering respiratory distress, pain or anxiety can cause for patients and their family [1,3,11].

Respiratory distress was the most common indication for initiating the OPD. Pain was, by contrast, much less often a trigger of OPD use. This contrasts with a general trend in the literature suggesting that pain is one of the most prevalent symptoms in pediatric palliative care [1,2,3,12]. Although our study focused on the symptoms of children with a life-threatening illness and at the end of life, the OPD was designed specifically to treat acute distress related to pain, dyspnea and anxiety. Our study was not concerned with the overall prevalence of these symptoms but focused on symptoms unresponsive to a patient’s usual medications. Many palliative care patients are prescribed opioids or other analgesics for pain; although they are occasionally prescribed anticholinergics for secretion management or benzodiazepines and opioids for breathlessness, the control and prevention of dyspnea might be more difficult than control and prevention of pain. Therefore, our findings support that intractable respiratory distress may be more common and more difficult to anticipate than other symptoms like an acute pain crisis in children receiving palliative care.

A common reason for initiating the OPD was gasping, which is a brainstem reflex unlikely to be affected by sedatives or analgesics. There are a few hypotheses as to why the OPD was administered for this reason: it reveals a need for educating healthcare professionals on the causes of gasping and to differentiate it from respiratory distress or dyspnea. It might also be that healthcare professionals administered the OPD as a safeguard measure to ensure any distress was controlled in patients gasping at the end of life, although gasping itself was not expected to cease following administration of the OPD.

There are a few reports of the efficacy of an order set in treating patients at the end of life, both to improve symptom control and to increase adherence to current standards in palliative care treatment [13,14,15]. In a pilot study of children at the end of life, Houlahan et al. have reported the use of rapid titration templates, among other strategies, to reduce the burden associated with rapid escalation of pain, dyspnea, and agitation [8]. The protocol used at Sainte-Justine Hospital is, according to the pediatric literature, the first one with an order set consisting of a medication regimen including three different classes of drugs (benzodiazepine, opioids and anticholinergic agents), and across routes of administration.

In this study, most patients died more than 24 h after the initial OPD administration. Despite the fact that literature is scarce on the pharmacokinetics and pharmacodynamics of drugs at the end of life, medications included in the OPD have a rapid onset of action and a peak effect within minutes to a few hours; the effects of these drugs are likely to be have decreased after a 24-h period, lending support to the safety of OPD use. The wide range of intervals between administration of the OPD and death likely reflects the unpredictable nature of both end of life and a patient’s illness trajectory.

Concerns related to the use of opioids or benzodiazepines at the end of life are pervasive, even among healthcare professionals, even though evidence supports their safety and effectiveness [16,17,18,19]. In a previous study in our institution on medical and nursing staff perceptions of the OPD, hastening death was a perceived disadvantage of the OPD [20]. Our study demonstrates that use of the OPD is safe, providing additional support for education of our community on its use in palliative care.

In our study, the interval between prescription of the OPD and its administration was longer in children with cancer than in children with other illnesses. One possible explanation is that healthcare professionals have a better understanding of the life trajectories of patients with cancer. Children with neurological disorders such as encephalopathy or cerebral palsy have a pattern of relatively stable condition with intermittent crises. The paucity of other life-threatening conditions precludes the determination of a typical life trajectory. Because the OPD is prescribed when there is reason to expect a palliative care patient might suffer from respiratory distress or acute pain, it might be easier for healthcare professionals to predict these occurrences in children with cancer. As a result, the OPD might have been prescribed earlier in the trajectory of oncology patients. In addition, half of the patients included in our study had a diagnosis of cancer. Progressive experience and comfort from the oncology team with the OPD as compared to other services might also have influenced the timing of its prescription.

There are several limitations to this study. First, this is a single center study; although specialized palliative care for children is often centralized in tertiary care institutions, our study reflects local consultation trends and practice customs, and might not be generalizable to other centers. Second, the standardized form used to guide selection of pharmacological agents excluded alternate routes of medication administration. Families and healthcare teams might have opted not to use the OPD given that placement of an IV line or an indwelling subcutaneous catheter were not concordant with the family’s goals of care. Notably, the sublingual route was added to a later iteration of the OPD prescription form. Third, alleviation of symptoms was determined in our study by reviewing nurses’ and physicians’ notes and the modified Borg pain scale. While this provided two ways to assess our protocol’s efficacy, data was missing for 22% of all OPD administration episodes. In addition, the modified Borg scale has not been validated in our patient population. To our knowledge, there are no validated tools for dyspnea in children at the end of life, and no scoring system targeting distress specifically. The choice of the modified Borg scale for our study was motivated by its previous use in other contexts at our hospital, ease of utilization, and because it was representative of the reality of patients with life-threatening illness or at the end of life. Fourth, nurses were responsible for scoring dyspnea in our study to optimize consistency of our data, and because many patients in our population (notably at the end of life) were not in a position to rate their dyspnea themselves. This being said, we could have allowed self-evaluation in cases where a patient, or a parent surrogate, was able to rate his or her distress. Additionally, we did not assess the family’s perception of the efficacy and relevance of our protocol. There might have been a discrepancy in evaluation of symptoms between healthcare professionals and the patient or parents. Lastly, we reviewed charts of patients for which the OPD was prescribed. As was noted by Walling et al., there probably exist patients in our center that would have benefited from the OPD but for whom the palliative care team was not consulted [21]. Further research into these missed opportunities would enlighten ways to improve delivery of high-quality palliative care.

## 5. Conclusions

At Sainte-Justine Hospital, the OPD was administered in 75% of patients for whom it was prescribed, mostly for respiratory distress. This study found that the OPD was effective in 90% of cases in which assessment data was available. In addition, the OPD was shown to be safe to use, both before and at the end of life.

In our study, the interval between prescription of the OPD and its administration was longer in children with cancer than in children with other illnesses. This may be in part due to our current, more advanced understanding of the life trajectories of patients with cancer.

Respiratory distress was by far the most common indication for initiating the OPD in our study. The OPD was designed to be used when symptoms are distressing and unresponsive to a patient’s usual medications, suggesting that intractable respiratory distress may be more common and more difficult to anticipate than other symptoms like an acute pain crisis in children receiving palliative care. Finally, gasping was a common reason for initiating the OPD. Further research to understand the attitudes and perceptions of professionals towards this symptom is needed.

## Figures and Tables

**Figure 1 healthcare-07-00003-f001:**
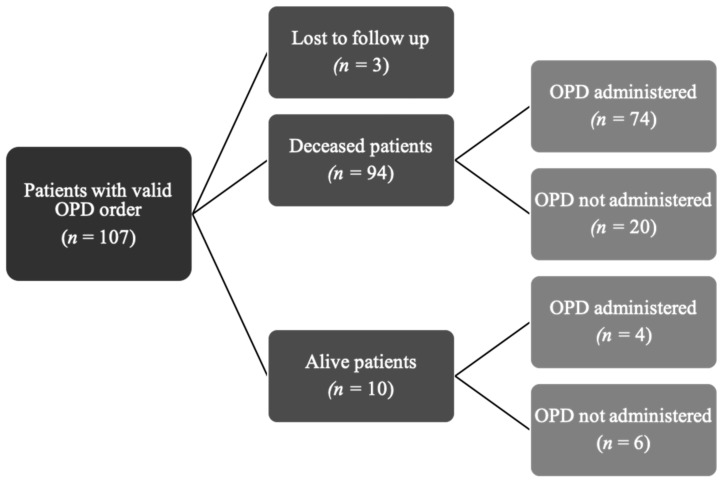
Patients with a valid OPD order at any time during the study period.

**Figure 2 healthcare-07-00003-f002:**
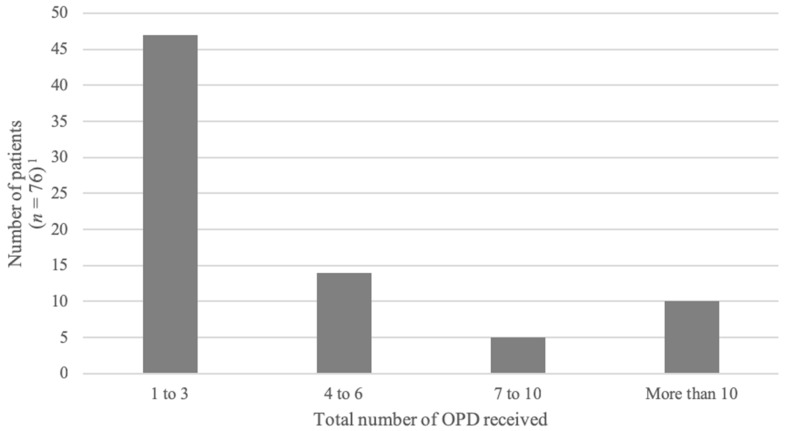
Total number of OPD received by each patient. ^1^ Data missing for 2 patients.

**Figure 3 healthcare-07-00003-f003:**
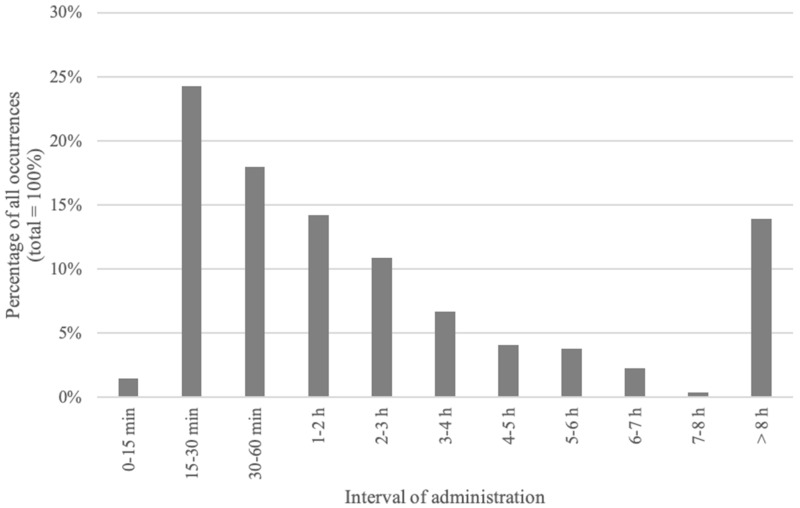
Interval between two OPD administration episodes.

**Figure 4 healthcare-07-00003-f004:**
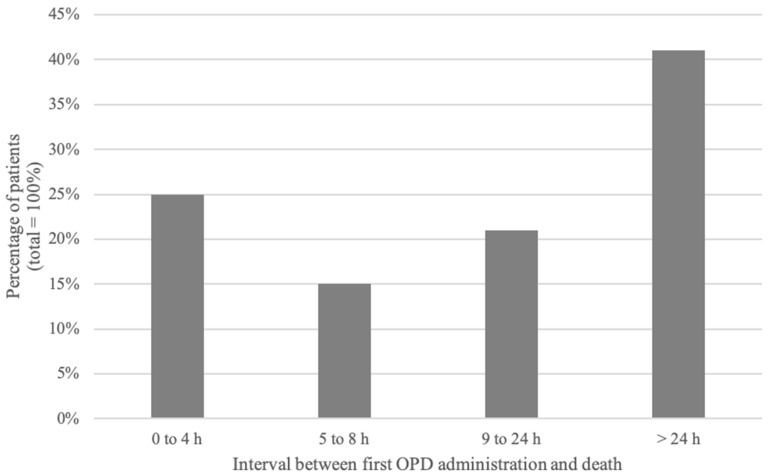
Interval between two OPD administration episodes.

**Table 1 healthcare-07-00003-t001:** Details of the Order Protocol for Distress (OPD).

OPD	Details
Inclusion criteria	The OPD is recommended by the palliative care team when judged appropriate if (a) there is a possibility of respiratory distress, acute pain crisis or anxiety, and (b) resuscitation status has been discussed and excludes the use of chest compressions, defibrillation and endotracheal intubation in the setting of clinical worsening or cardiopulmonary arrest.
Discussion with family	There is a discussion between the physician and the patient’s parents (and patient when appropriate) to explain the OPD (indication, medication used, anticipated effects). The OPD is prescribed if parents (and patient when appropriate) agree with the protocol.
Prescription	After agreement from the family, OPD medications are prescribed by a physician according to recommended doses. OPD consists of (1) midazolam (0.1 mg/kg if naive or 0.15 mg/kg if already on benzodiazepines), with (2) morphine (0.1 mg/kg if naive or 10% to 20% of daily morphine dose) or hydromorphone (0.02 mg/kg if naive or 10% to 20% of daily dose), with or without (3) glycopyrrolate (0.004 mg/kg) or scopolamine (0.006 mg/kg). Prefilled syringes are kept in a locked cabinet on the care unit accessible by the patient’s bedside nurse.
OPD initiation	The bedside nurse can initiate the OPD whenever the patient presents severe respiratory distress, acute pain or anxiety that is not relieved by usual measures; the nurse must notify the physician as soon as the OPD is initiated. The medication is administered only if parents (or the patient when appropriate) agree with the use of the protocol. If the parents refuse, the nurse notifies the physician and waits before the initiating the OPD; all other relevant medication or comfort measures are maintained.
Evaluation by the physician	Physicians are required to come to the patient’s bedside whenever the OPD is initiated to evaluate the cause of distress, adjust medication and provide support to the patient and family.
Repetition of the OPD	Nurses can repeat administration of the OPD every 15 min and up to two times, as necessary, until the physician arrives at the bedside.
Out of hospital providers	The medication can be used at home by community nurses or by parents (after evaluation and training by the palliative care team). Medications can be prepared by the hospital pharmacist or the patient’s local pharmacist. They are stored in a separate kit at home in prefilled, labeled syringes with a guide for utilization (to minimize dosing or administration errors).

**Table 2 healthcare-07-00003-t002:** Patients characteristics (*n* = 104).

Variables	Characteristics	Number of Patients (%)
Gender	Male	45 (43)
	Female	59 (57)
Diagnosis	Cancer	50 (48)
	Cerebral tumor	17 (16)
	Leukemia/lymphoma	12 (12)
	Bone/muscle/tissue tumor	14 (13)
	Neuroblastoma	4 (4)
	Wilms/liver tumor	3 (3)
	Neurologic disorders	29 (28)
	Encephalopathy (incl. anoxic brain injury and seizure disorders)	24 (23)
	Spinal muscular atrophy type 1	5 (5)
	Genetic disorders	11 (11)
	Congenital syndrome	9 (9)
	Inborn errors of metabolism	2 (2)
	Cardiac disease	9 (9)
	Congenital cardiopathy	7 (7)
	Others	2 (2)
	Pulmonary disease	3 (3)
	Others	2 (2)

Patients’ age (years): 7.1 (range 0.04–22).

**Table 3 healthcare-07-00003-t003:** Characteristics of the OPD administered (*n* = 350).

**Medication**	**Dose mg/kg Median (Range)**	**Number of OPD (%)**
Benzodiazepine		
Midazolam	0.1 (0.03–0.4)	344 (98)
Opioid		
Morphine	0.1 (0.04–0.6)	261 (75)
Hydromorphone	0.1 (0.02–0.35)	83 (24)
Anticholinergic agent		
Glycopyrrolate	0.004 (0.003–0.005)	89 (25)
Scopolamine	0.006 (0.002–0.008)	23 (7)
**Place of ODP administration**	**Number of OPD (%)**	
Sainte-Justine Hospital	239 (68)	
Lighthouse Hospice	74 (21)	
Marie-Enfant Rehabilitation Center	25 (7)	
Home	12 (3)	
**Route of administration**	**Number of OPD (%)**	
Subcutaneous	184 (53)	
Intravenous	166 (47)	
**Person who administrated the OPD**	**Number of OPD (%)**	
Nurse	261 (75)	
Attending physician	57 (16)	
Resident	21 (6)	
Parents	10 (3)	
Other—not specified	1 (0.3)	
**Indication stated for administration**	**Number of OPD (%)**	
Respiratory distress	310 (89)	
Pain crisis	29 (8)	
Anxiety	4 (1)	
Other	4 (1)	
Unknown	3 (1)	

**Table 4 healthcare-07-00003-t004:** Effectiveness of the OPD to relieve symptoms (*n* = 273) ^1^.

Relief	Number of Episodes (%)
Relief and partial relief	245 (90)
Effective relief	185 (68)
Partial relief	60 (22)
No relief	28 (10)

^1^ Data missing for 77 of 350 total episodes.

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
