# Peer review of "Clinical Use of an Order Protocol for Distress in Pediatric Palliative Care"

_healthcare, 2019, doi:10.3390/healthcare7010003_

Round 1

Reviewer 1 Report

Thank you for this well-written paper. I very much enjoyed reading about your study. I did have two comments which I wished to share with you:

1) You are using the phrase, 'they were lost to follow-up' to refer to participants who probably did not engage with the study after the initial discussions? I find the phrase oddly worded and peculiar. Can it be reworded, for accuracy and clarity?

2) The conclusions are brief and underdeveloped. There surely is something more to conclude with?

Author Response

Response to Reviewer 1 comments

Point (1): You are using the phrase, 'they were lost to follow-up' to refer to participants who probably did not engage with the study after the initial discussions? I find the phrase oddly worded and peculiar. Can it be reworded, for accuracy and clarity?

Response (1): Thank you for this comment.

By using ‘lost to follow-up’, we did not refer to families who refused to consent. Given that this study is a retrospective chart review, the discussion, consent and implementation of the OPD had already been done with families.

This term was used to designate patients whose charts became unavailable or inexistent – usually because of a transfer to another hospital. These patients stopped being followed at our hospital center, hence our use of their phrase, ‘lost to follow-up’. 

We amended our article accordingly (see Results section).

Point (2): The conclusions are brief and underdeveloped. There surely is something more to conclude with?

Response (2): We have expanded our conclusion to include a summary of several more elements of our study. We have highlighted the two major outcomes of our study (safety and effectiveness of the OPD) and mentioned that the OPD was used in 75% of patients for whom it was prescribed.

We have also reiterated the high frequency of respiratory distress as an indication for the OPD administration, and our finding that gasping was a common target for the OPD. We also added a note on the interval between prescription of the OPD and its administration, which was longer in children with cancer than in children with other illnesses.

Reviewer 2 Report

A well written paper.  This study has contributed to the international literature.

Could the authors please elaborate on the benefits of treating all symptoms as distress compared to singling out the symptoms of pain, anxiety and respiratory distress?

How was the Borg scale implemented?  Did the patients rate themselves?  If so, how many patients were too young or cognitively affected so as to not be able to do this.  Why were pain scales not used, in addition to the Borg scale?  Has the Borg scale been validated for use amongst children?

Can the authors comment on how medications are stored at home, if possible, please?

Further emphasis of the OPD being used to treat the distressing symptoms of a patient as they are dying, as opposed to OPD causing or triggering OPD could be made.  The authors do this, but I feel it could be highlighted more.

Can the authors comment on why respiratory distress was the predominant symptom managed with the OPD, when research suggests that pain is possibly the most prevalent symptom at end of life.

The authors make a good point on the role of education in implementing clinical use of OPD.

The challenge of managing gasping was also described.  This could be an area of further study.

The authors were able to discuss some of their findings very well, including the differences between patients with oncology and non-oncology diagnoses.

The conclusion was very brief, and it could be helpful to add one or two other key findings from the study to the conclusion.

Author Response

Response to Reviewer 2 comments

Point (1): Could the authors please elaborate on the benefits of treating all symptoms as distress compared to singling out the symptoms of pain, anxiety and respiratory distress?

Response (1): Our goal was to create a tool that would enable rapid and effective treatment of some of the most common symptoms experienced by pediatric palliative care patients. By using a standard order aimed at pain, anxiety and respiratory distress, we were hoping to streamline the administration of medications to alleviate these symptoms as early as possible.

This being said, the OPD was not created to address all symptoms at the end of life. For instance, a separate order set was created at our hospital to address terminal bleeding episodes (this is not mentioned in the article under review); similarly, seizures at the end of life are addressed separately and are not a target symptom for the OPD. We have reworded our description of the OPD in the article to avoid any confusion on the scope of its indications.

We have added this clarification at the beginning of our article’s “Discussion”.

Point (2): How was the Borg scale implemented?  Did the patients rate themselves?  If so, how many patients were too young or cognitively affected so as to not be able to do this.  Why were pain scales not used, in addition to the Borg scale?  Has the Borg scale been validated for use amongst children?

Response (2): These are important questions and we thank you for raising them.

A passage was added in our article to describe how the Borg scale was implemented: “Given there is no validated scale for dyspnea in the pediatric palliative care population, the modified Borg Scale (1 to 10) was used in this study. To ensure uniformity, nurses were responsible for recording the score in every case; the same nurse would rate the patient’s dyspnea both before and after administration of the OPD medications.” We decided that the nursing staff would be responsible for scoring dyspnea in our study also in part because many patients in the pediatric palliative care population (notably at the end of life) were not in a position to rate their dyspnea themselves. This being said, we have added this as a limitation in our discussion.

The Borg scale was identified as our tool to measure distress for two reasons: first, we had evidence from our own clinical use of the OPD that the great majority of episodes of distress were related to dyspnea. Second, because there is no tool measuring distress from symptoms in a given population, we used the scoring system that was likely to be used in most instances.

To our knowledge, there are no validated tools for dyspnea in children at the end of life. The Borg scale has been validated in a few pediatric populations and settings. It has, for instance, been shown to be valid in children with cystic fibrosis to measure the sensation of dyspnea after physical activity. The choice of the Borg scale for our study was motivated by its previous use in other contexts at our hospital and relative ease of utilization. In addition, several pediatric dyspnea scores exist and are used for asthma: Pediatric Dyspnea Scale (PDS), Asthma Score (AS), Asthma Severity Score (ASS), Clinical Asthma Evaluation Score 2 (CAES-2), Pediatric Respiratory Assessment Measure (PRAM) and respiratory rate, accessory muscle use, decreased breath sounds (RAD). Among these tools, only the PRAM score has been validated, but for the assessment of acute asthma exacerbation, which we considered not transferable to the pediatric palliative care population. Hence, because there are no validated tools for the evaluation of dyspnea in the pediatric palliative care, we decided to select a score that was common in adult medicine, had a well-designed modified version for pediatrics, and seemed closest to the reality of patients with life-threatening illness or at the end of life.

Accordingly, we have added the absence of a validated tool as a limit in our discussion, alongside our earlier comment on having nurses fill the score every time (as opposed to allowing the patient or a parent to score whenever possible).

Point (3): Can the authors comment on how medications are stored at home, if possible, please?

Response (3): We did not collect data on medication storage as part of our study. This being said, it is common practice at our hospital for “emergency medications” (such as those used in the OPD) to be stored in a separate kit at home, in prefilled, labeled syringes with a guide for utilization (to minimize dosing or administration errors). Medications can be prepared by the hospital pharmacist or the patient’s local pharmacist.

This was clarified in our article’s Table 1.

Point (4): Further emphasis of the OPD being used to treat the distressing symptoms of a patient as they are dying, as opposed to OPD causing or triggering OPD could be made.  The authors do this, but I feel it could be highlighted more.

Response (4): We are unsure about the question above but understand it to mean that more emphasis should be made on the fact that the OPD relieved distress at the end of life, rather than the OPD causing distress or precipitating death.

We have added the following comment in our discussion to emphasize this point: “Despite the fact that literature is scarce on the pharmacokinetics and pharmacodynamics of drugs at the end of life, medications included in the OPD have a rapid onset of action and a peak effect within minutes to a few hours; the effects of these drugs are likely to be have decreased after a 24-hour period, lending support to the safety of OPD use.”

Point (5): Can the authors comment on why respiratory distress was the predominant symptom managed with the OPD, when research suggests that pain is possibly the most prevalent symptom at end of life.

Response (5): Thank you for this important remark.

We will emphasize this point in an updated version of our discussion. Indeed, the literature suggests that pain might the most prevalent symptom at the end of life, or at least as common as dyspnea in children with life-threatening illnesses. For instance, Wolfe’s article in the New England Journal of Medicine found that children with cancer experienced dyspnea almost as often as pain.

Our study is the first, to our knowledge, to investigate the use of an OPD to treat acute distressthat is not responsive to a patient’s other medications. Thus, the OPD is commonly used for unexpected rapidly progressing symptoms. Many palliative care patients have opioids or other analgesics directed at alleviating discomfort. Although patients often be prescribed anticholinergics for secretion management, or benzodiazepines and opioids for breathlessness, the control and prevention of dyspnea might be more difficult than that of pain.

Our study did not address the prevalence of pain or dyspnea overall in the patient population, but of acute distress related to pain, dyspnea or anxiety unrelievedby a patient’s usual medications. Hence, our study might have helped uncovering that intractable respiratory distress is more difficult to treat than intractable pain in children receiving palliative care – an element that would be worth exploring in further studies.

Point (6): The challenge of managing gasping was also described.  This could be an area of further study.

Response (6): We have included this comment in an expanded version of our conclusion.

Point (7): The conclusion was very brief, and it could be helpful to add one or two other key findings from the study to the conclusion.

Response (7): We have expanded our conclusion to include a summary of several more elements of our study. We have highlighted the two major outcomes of our study (safety and effectiveness of the OPD) and mentioned that the OPD was used in 75% of patients for whom it was prescribed.

We have also reiterated the high frequency of respiratory distress as an indication for the OPD administration, and our finding that gasping was a common target for the OPD. We also added a note on the interval between prescription of the OPD and its administration, which was longer in children with cancer than in children with other illnesses.